# Defect-Enriched Graphene Nanoribbons Tune the Adsorption Behavior of the Mediator to Boost the Lactate/Oxygen Biofuel Cell

**DOI:** 10.3390/nano13061089

**Published:** 2023-03-17

**Authors:** Xiaoyu Feng, Yongyue Ning, Zhongdong Wu, Zihan Li, Cuixing Xu, Gangyong Li, Zongqian Hu

**Affiliations:** 1College of Textiles and Clothing, Xinjiang University, Urumqi 830046, China; 2Beijing Institute of Radiation Medicine, Beijing 100850, China; 3Key Laboratory of Nanobiosensing and Nanobioanalysis, Universities of Jilin Province, Northeast Normal University, Changchun 130024, China; 4Key Laboratory of Hunan Province for Advanced Carbon-Based Functional Materials, School of Chemistry and Chemical Engineering, Hunan Institute of Science and Technology, Yueyang 414006, China

**Keywords:** enzymatic biofuel cell, bioelectrocatalysis, graphene nanoribbon, energy conversion

## Abstract

Owing to the high efficiency and specificity in moderate conditions, enzymatic biofuel cells (EBFCs) have gained significant interest as a promising energy source for wearable devices. However, the instability of the bioelectrode and the lack of efficient electrical communication between the enzymes and electrodes are the main obstacles. Herein, defect-enriched 3D graphene nanoribbons (GNRs) frameworks are fabricated by unzipping multiwall carbon nanotubes, followed by thermal annealing. It is found that defective carbon shows stronger adsorption energy towards the polar mediators than the pristine carbon, which is beneficial to improving the stability of the bioelectrodes. Consequently, the EBFCs equipped with the GNRs exhibit a significantly enhanced bioelectrocatalytic performance and operational stability, delivering an open-circuit voltage and power density of 0.62 V, 70.7 μW/cm^2^, and 0.58 V, 18.6 μW/cm^2^ in phosphate buffer solution and artificial tear, respectively, which represent the high levels among the reported literature. This work provides a design principle according to which defective carbon materials could be more suitable for the immobilization of biocatalytic components in the application of EBFCs.

## 1. Introduction

EBFCs have been recognized as a promising power source for various wearable electronic devices owing to the use of cheap and renewable biocatalysts, the utilization of abundantly available high-energy-density biofuels, and their capability to operate under mild temperature and pH conditions [1,2,3]. Currently, the output power of EBFCs is limited by the lack of efficient electrical communication between the enzymes and electrodes due to the active center of enzymes being embedded in the inner shell of the protein [4,5]. Mediated electron transfer (MET) using redox mediators to shuttle electrons between enzymes and electrodes has been proposed to overcome the critical problem of poor electrical communication [6,7,8,9]. MET can usually bring higher catalytic current, which, to some extent, compensates for the insufficient power density reduction in EBFCs caused by the potential loss [10]. Therefore, the stable immobilization of the mediators plays a pivotal role in the MET-based EBFC systems because the leakage of mediators from the electrode into the electrolyte would deteriorate the stability of the EBFCs [11,12,13,14].

For another, in order to improve the output power of EBFCs, it is necessary to immobilize the enzymes and the mediators to increase the effective mass transfer of fuel on the electrode surface. The nanostructured carbon materials can provide a good confinement environment for enzyme immobilization and promote efficient electron transfer, making the research on carbon-based EBFCs flourish [15,16]. In this regard, several carbonaceous materials such as graphene and carbon nanotubes (CNTs) have been utilized as Appendix A for the immobilization of enzymes [17]. Although the above-mentioned carbon materials have been widely used in EBFCs, their non-polar characteristics and poor affinity towards polar mediators result in the poor electrode stability [18]. The instability of the bioelectrode is the main obstacle that limits the performance of the current EBFCs. Therefore, the rational design of Appendix A with polar characteristics to enhance the affinity towards the polar mediators and revealing the interaction mechanism between them are of great significance to guiding the design of electrode materials and overcoming the performance bottleneck of EBFCs in practical applications [12,19,20].

Graphene nanoribbons are narrow, lengthened strips of graphene that have a high length-to-width ratio [21]. GNRs with rich edge sites have been developed via the longitudinal unzipping of CNTs in an oxidative acid solution [22]. Unlike the pristine CNTs, GNRs have a high surface area and abundant electrochemically active edge sites, thus rendering them intriguing materials for use in energy conversion and storage [23,24,25] and especially in biomedicine and biosensor applications due to their good adsorption capability and biocompatibility [26,27]. However, little attention has been paid to the application of GNRs in the field of EBFCs.

Herein, we demonstrate the first example of the defect-enriched graphene nanoribbons being applied to membrane-less lactate/oxygen biofuel cells to harvest the electrical power from a low concentration of lactic acid in tears. The three-dimensional defect-enriched graphene nanoribbons (GNRs) frameworks are rationally designed and synthesized by longitudinally unzipping the sidewalls of multi-walled CNTs, followed by hydrothermal N-doping and thermal annealing. The obtained GNRs are used as Appendix A to immobilize enzymes and mediators, showing enhanced catalytic performance and stability compared with conventional graphene and CNTs. Density function theory (DFT) calculation reveals that the adsorption energy of the defective carbon to the representative mediator tetrathiafulvalene (TTF) is 0.84 eV, which is much larger than the adsorption energy of pristine carbon to the TTF (0.073 eV). Finally, the assembled membrane-less lactate/O_2_ biofuel cell using the resulting GNRs as the supporting electrode delivers an open-circuit voltage (OCV) of 0.63 V and a maximum power density of 70.7 μW/cm^2^ in 0.1 M phosphate buffer solution (PBS) and exhibits an OCV of 0.65 V and a maximum power density of 18.6 μW/cm^2^ in an artificial tear. The latter shows a better performance than the previously reported lactate/O_2_ biofuel cells in tears with the same concentration of lactate (Appendix A). In addition, a long-term continuous discharge for ca. 2.3 h is achieved at 0.02 mA/cm^2^. Therefore, the as-synthesized GNRs are promising in wearable bioelectronics applications.

## 2. Experimental Section

### 2.1. Chemicals and Materials

Lactate oxidase (LOx, 42 U/mg solid, from Microorganism), bilirubin oxidase (BOD, 37 U/mg solid, from Myrothecium verrucaria) and TTF (Sigma-Aldrich Inc., St. Louis, MO, USA). 2,2′-diazide-bis-3-ethylbenzothiazolin-6-sulfonic acid (ABTS, Aladdin Inc., Shanghai, China) and lactate (80%, Aladdin Inc., Shanghai, China). Chitosan solution (Aladdin Inc., Shanghai, China), bovine albumin (BSA, Aladdin Inc., Shanghai, China), and Multiwall CNTs (Appendix A) were purchased from Xuzhou Jiechuang New Material Technology Co., Ltd. (Jiangsu, China). The glassy carbon electrode was purchased from Tianjin Aida Hengsheng Technology Development Co., Ltd. (Tianjin, China). The synthetic tear solution was composed of 150 mM PBS (pH 7.2), 0.05 mM β-D-glucose, 3 mM L-lactate, 0.18 mM L-ascorbate, 5.4 mM urea, 2.47 mg/mL lysozyme, 0.2 mg/mL BSA, and 0.15 mg/mL mucin. The pH of the mixture was adjusted to 7.2 using 0.1 M NaOH.

### 2.2. Synthesis of Defect-Enriched GNRs

Typically, 1 g of CNTs was suspended in 30 mL of concentrated H_2_SO_4_ and then stirred at room temperature for 12 h. A total of 5 g of KMnO_4_ was gradually added into the mixture with an ice water bath and stirred at room temperature for 1 h. The mixture was moved to an oil bath at 65 °C for 30 min. After that, the solution was quenched in ice water, and 100 mL of ice water and 5 mL of H_2_O_2_ (30 wt%) were poured slowly into the mixture. The mixture was centrifuged and washed with HCl (1/10, *v*/*v*) and deionized water. The product was lyophilized to obtain the precursor. A total of 0.3 g of the obtained precursor and 9 g of urea were dispersed in 50 mL of deionized water. After sonication and stirring for 30 min, the suspension was transferred into a 200 mL autoclave and heated at 180 °C for 12 h. After cooling down to room temperature, the mixture was washed with deionized water and dried at 60 °C to obtain O-GNRs. Finally, the O-GNRs composite was thermally annealed at 1600 °C for 2 h in a tube furnace under the protection of Ar to obtain GNRs. A schematic diagram of the preparation procedures of the GNRs composite is shown in Figure 1a.

### 2.3. Preparation of the LOx/TTF/GNRs/GCE Bioanode

A glassy carbon electrode (GCE) was sequentially polished using a slurry of 0.5 μm and 0.03 μm alumina power and successively washed by ultrasonication in deionized water, ethanol, and deionized water for 3 min, respectively. Then, the electrode was activated electrochemically in 0.1 M KCl containing 5 mM [Fe(CN)_6_]^3−/4−^ aqueous solution by cyclic voltammetry (CV) until the CV curves were invariant. Thereafter, the electrode was washed with deionized water and dried under infrared light illumination. For the preparation of the bioanode, 1 μL of BSA (10 mg/mL, in 0.1 M PBS (pH 7.2)), 2 μL of ethanol solution containing 30 mM TTF, 4 μL LOx (40 mg/mL, in 0.1 M PBS (pH 7.2)), and 4 μL GNRs (5 mg/mL, in N,N-dimethylformamide) were mixed. After that, 5 μL of the above mixture was dropped on the cleaned GCE and then dried at room temperature. The obtained electrode was denoted as LOx/TTF/GNRs/GCE. After drying at room temperature, 2 μL of a Chit solution (1 wt%, in glacial acetic acid) was cast on the electrode to act as a protective and biocompatible layer.

### 2.4. Preparation of the BOD/ABTS/GNRs/GCE Biocathode

For the preparation of the biocathode, 0.5 mL of 0.1 M PBS (pH 7.2) was loaded in a clean centrifuge tube; a mixture of GNRs (2.5 mg) and ABTS (0.49 mg) was then added to the PBS and left to stand for two days [28]. Subsequently, the mixture was centrifuged 2–3 times, and 0.5 mL of PBS was added. Finally, 5 μL of the mixture was cast on the GCE and dried under infrared light illumination. After drying, 5 μL of BOD (10 mg/mL, in 0.1 M PBS) was loaded and dried at room temperature. Finally, 2 μL of Nafion solution in ethanol (1 wt%) was cast on the electrode to act as a protective and biocompatible layer.

### 2.5. Electrochemical Measurements

Electrochemical experiments were conducted on an electrochemical workstation (CHI 660E, CHI Instrument, Shanghai, China). CV measurements of the LOx/TTF/GNRs/GCE bioanode and BOD/ABTS/GNRs/GCE biocathode were performed under a typical three-electrode system using Ag/AgCl (3 M KCl) as a reference electrode and Pt foil (1 cm^2^) as a counter electrode in 0.1 M PBS (pH 7.2) at a scan rate of 10 mV/s. CV measurements on BOD/ABTS/GNRs/GCE were carried out in 0.1 M PBS (pH 7.2) with or without free 0.5 mM ABTS as a control experiment for oxygen reduction. The stability experiment was operated intermittently. First, the CV curve was detected in the presence of lactate at a scanning rate of 10 mV/s in 0.1 M PBS (pH 7.2); then, the electrode was gently removed, rinsed with deionized water, and stored in the refrigerator at 4 °C. The operation was repeated every day for a 5-day CV test. All experiments were performed at ambient temperature.

The lactate/O_2_ EBFC was fabricated in a single compartment. The LOx/TTF/GNRs/GCE and the BOD/ABTS/GNRs/GCE acted as the bioanode and the biocathode, respectively. Further, 0.1 M PBS (pH 7.2) containing 3 mM lactate and the synthetic tear solution were used as the electrolyte.

### 2.6. Characterizations

X-ray diffraction (XRD) was performed on the PANalytical X’pert Pro X-ray diffractometer. The Raman spectrum was recorded on a WITecCRM200 instrument using a 532 nm laser. The field emission scanning electron microscope (FESEM) was operated on a JSM-6701 field emission SEM instrument with an acceleration voltage of 10 kV. A transmission electron microscope (TEM) was performed on a JEOL JEM-2100 microscope, which was operated at an acceleration voltage of 200 kV. X-ray photoelectron spectroscopy (XPS) was carried out on a Thermo Scientific Escalab 250 spectrometer using an ALA transmission Kα X-ray source. The N_2_ adsorption measurement was performed at 77 K on the Micromeritics ASAP 2020 HD analyzer. Fourier transform infrared spectroscopy (FT-IR) was recorded on the Spectrum 100 FT-IR. Atomic force microscopy (AFM) was performed on a Bioscope Catalyst. The Gaussian program (Gaussian) was adopted to realize the DFT to evaluate the exchange of electrons and related effects and calculate the molecular energy and structure.

### 2.7. Computational Methods and Models

Density function theory (DFT) calculations were based on the Gaussian 16 software package [29]. Under the B3lyp functional, the 6–31 G (d, p) basis set was used for the geometric optimization of the model. The adsorption energy was defined by
Eads = E(surf + mol) − E(surf) − E(mol)
where E(surf + mol) was the total energy of a surface interacting with a molecule, E(surf) was the energy for graphene or defect graphene, and E(mol) was the energy for TTF.

## 3. Results and Discussion

### 3.1. Characterization of the GNRs

Appendix A shows the SEM images of the CNTs. CNTs are composed of cross-linked one-dimensional nanostructures with a diameter from 30 to 100 nm. The TEM images in Appendix A reveal that the original CNTs are typical one-dimensional nanostructures. After the chemical oxidation and hydrothermal process, the O-GNRs show a fluffy porous network morphology with a partial unzipped structure (Appendix A). The width of the carbon nanostructure significantly increases after the unzipping and thermal treatment process (Figure 1b). The selected area electron diffraction (SAED) pattern shows a bright diffraction ring (Figure 1c), indicating a high degree of graphitization for GNRs. As shown in the high-resolution TEM (HRTEM) image (Figure 1d), the nanotubes are cut into several layers to form a ribbon structure with a distance of 0.34 nm between adjacent graphite layers, which is characteristic of crystalline graphite [30]. The results suggest that the sidewalls of the CNTs can be completely unfolded to produce GNRs accompanied by a large number of edge carbon defects. The SEM and TEM images show that GNRs exhibit a three-dimensional cross-linked networked structure. Figure 1e,f show the AFM images of GNRs, which reveal a ribbon-like morphology with a stacking thickness ranging from tens to hundreds of nanometers. The 3D AFM patterns show that the surface of GNRs is rough (Figure 1g), which is beneficial for the immobilization of enzymes.

Raman spectroscopy was commonly used as an important technique for evaluating the structural information of carbon materials. As shown in Figure 2a, there are two wide bands at 1360 cm^−1^ (D band) and 1598 cm^−1^ (G band). The G-band and D-band show the in-plane vibration modes of sp^2^-bonded carbon atoms and sp^3^-bonded carbon atoms, respectively. The D-band reflects disorders and defects in the graphite lattice, while the G-band corresponds to ordered graphite-like sp^2^ carbon atoms [31]. Note that the I_D_/I_G_ peak intensity ratios of CNTs, O-GNRs, and GNRs are 0.95, 1.16, and 1.0692, respectively, indicating that hydrothermal treatment and subsequent annealing could introduce defects in the material.

The crystal structures of CNTs, O-GNRs, and GNRs were evaluated by XRD measurement. As shown in Figure 2b, we observed two main peaks at 26° and 43° that belong to C (JCPDS card No. 75-1621), corresponding to the (002) and (100) crystal planes of hexagonal graphite [17]. After hydrothermal treatment, the half-width of O-GNRs narrows at 26°, and the diffraction peaks are evidently shifted to the low angle, indicating that the interlayer spacing of the material becomes larger after heat treatment, thereby reducing the degree of the graphitization domain of the materials. After annealing at 1600 °C, the half-value width of GNRs returns to the initial level, demonstrating that the annealing treatment increases the graphitization degree of GNRs, which is beneficial to improving the electric conductivity of the material. The graphitization change can be explained as follows: Hydrothermal treatment with urea could introduce N and O species into the carbon framework of O-GNRs, which further increases the defect degree, thereby enlarging the interlayer spacing of (002) planes. After annealing at 1600 °C, the structure of GNRs can be partially recovered back to CNTs due to the combustion of N and O species under a high temperature, which results in more defects in GNRs than those of CNTs.

The structure of CNTs, O-GNRs, and GNRs was further investigated by FT-IR. As shown in Figure 2c, the peaks at around 3400, 2839, 1769, 1121, and 1055 cm^−1^ correspond to the stretching vibration of the O-H, -CH_2_−, C=O, C-OH, and C-O groups on O-GNRs, respectively [32]. Additionally, two new peaks at 1400 and 1250 cm^−1^, corresponding to C=N and C-N stretching vibration, are detected, indicating the doping of N atoms into the carbon framework after hydrothermal treatment. Notably, the peaks of the functional groups are weakened or disappear after thermal treatment at 1600 °C, indicating that the structure of GNRs changes after high-temperature annealing, and the oxygen-containing functional groups are significantly decreased after heat treatment, which is beneficial to increasing the electric conductivity of the GNRs.

Figure 2d shows the wide scan XPS spectra of the products. C and O elements exist in CNTs. After unzipping and hydrothermal treating with urea, N species are present in O-GNRs, and the content of O 1s is significantly increased. After annealing at 1600 °C, the N 1s peak disappears, and the O 1s peak is significantly weakened, which is consistent with the FT-IR results. Appendix A depicts the XPS spectra of C 1s and O 1s of CNTs, O-GNRs, and GNRs samples, respectively. The results reveal that thermal annealing could significantly decrease the O and N contents in the carbon materials.

Figure 2e shows the high-resolution N 1s XPS spectra of the samples. No N species can be detected in pristine CNTs. Pyrrolic-N (N-5, ~400.2 eV), pyridinic-N (N-6, ~398.5 eV), quaternary-N (N-Q, ~401.5 eV), and pyridinic-N-oxide (N-X, ~402.8 eV) [33] peaks appeared after hydrothermal treatment with urea, demonstrating the successful N-doping into the carbon framework. After further annealing at 1600 °C, the N-containing functional groups disappeared. The removal of N and O species during the annealing process could induce vacancies within the carbon framework, resulting in a defect-enriched carbon material.

The surface area and pore structure of the CNTs and GNRs composite were evaluated by N_2_ isotherm adsorption–desorption experiments. As shown in Figure 2f, according to the IUPAC classification, the isotherm shows type IV adsorption–desorption behavior and has a large hysteresis loop in the relative pressure range of 0.85 to 1.0, which indicates the presence of mesopores and macropores. The specific surface area (SSA) and pore volume of these materials were estimated using the Brunauer–Emmett–Teller (BET) method and the Barrett–Joyner–Halenda (BJH) model, respectively. The SSAs of GNRs and CNTs are 318.69 m^2^/g and 218.59 m^2^/g, and the pore volumes are 0.56 cm^3^/g and 0.32 cm^3^/g, respectively. The results show that the SSA of GNRs can be enlarged by unzipping CNTs, followed by high-temperature annealing. The enhanced SSA and pore volume could increase the mass loading of enzymes and mediators, thus increasing the catalytic currents.

### 3.2. Electrochemical Performance of the Bioelectrodes

The electrochemical performance of bare GCE and GNRs/GCE was investigated by CV 0.1 M KCl containing 5 mM [Fe (CN)_6_]^3−/4−^. As shown in Figure 3a, the peak current of the GNRs/GCE is higher than that of GCE, indicating that GNRs/GCE has a larger electrochemical active surface area than bare GCE. The peak-to-peak difference of GCE is larger than that of GNRs/GCE, demonstrating the weaker polarization and better reversibility of the GNRs. These results indicate the feasibility of GNRs as a supporting electrode for EBFCs. Appendix A show CV curves of CNTs/GCE and GNRs/GCE at different scan rates of 2–12 mV/s. According to the relationship of j vs. v (Appendix A), the double-layer capacitance (Cdl) of GNRs/GCE is larger than that of CNTs/GCE. The increased Cdl of GNRs/GCE could be mainly ascribed to the enhanced SSA and pore volume, which is consistent with the BET test results.

The electrochemical performance of graphene, CNTs, and GNRs loaded with LOx and TTF was compared using the CV measurements. As shown in Appendix A, yellow substances leaking from the electrode during CV measurements are observed in graphene- and CNTs-functionalized electrodes, while no changes can be observed in GNRs-functionalized electrodes (Appendix A). The current densities of graphene- and CNTs-modified electrodes are drastically decreased after 50 cycles (Appendix A), while the GNRs-functionalized electrode can maintain ca. 90% of its initial current density after 50 cycles (Appendix A). Appendix A shows the CV curve of bare GCE in the graphene-functionalized electrode-cycled electrolyte. A pair of redox peaks at the potential range of 0.1–0.3 V (vs. Ag/AgCl) can be detected, corresponding to the redox of the mediator TTF (Appendix A). Therefore, the leakage of the substances in the electrolyte is attributed to the mediator TTF. The result indicates that GNRs can suppress the elution of TTF to some extent.

DFT calculation was then carried out to investigate the interaction of the carbon materials and the mediator TTF. The structure of the pristine graphene and defective graphene used in the DFT calculations is shown in Appendix A. The results are shown in Figure 3b; the adsorption energy between the defective carbon and TTF is 0.84 eV, which is much higher than that of the pristine carbon and TTF (0.073 eV), indicating that the defective carbon has a stronger adsorption ability for TTF than pristine carbon. The enhanced adsorption between the GNR and the mediators can be attributed to the polarity that presents in both the mediators and the defective GNRs. It should be noted that mediators such as TTF and ABTS are both polar organic molecules. The introduction of defects into the perfect carbon framework could inevitably cause structural and electronic distortions, leading to alterations in the charge transport, Fermi level, bandgap, localized electronic state, and spin density [32,33], thus showing a much stronger adsorption property towards polar molecules than pristine carbon. The result further reveals that TTF adsorbed on the surface of the GNRs electrode can form a stable architecture for effectively mediating the electron transfer between LOx and the electrode. In addition, the strong adsorption effects could shorten the electron transfer distance and ensure fast electron transfer kinetics, thus improving the catalytic current. Both enhanced electrode stability and fast electron transfer kinetics are beneficial to MET.

The catalytic current density of bioanode mainly depends on the process of interfacial electron transfer and biocatalytic reaction. When GCE was modified by the Appendix A, enzyme, and mediator, the catalytic current of the bioanode would be affected by the concentration of the enzyme and mediator at a certain amount of Appendix A. The oxidation of lactate on the bioanode depends on the bioelectrocatalytic process of the enzyme and mediator. Therefore, the electrocatalysis performance of the bioanode with the immobilization of different amounts of mediators and enzymes was systematically studied. As shown in Appendix A, when the amount of enzyme modification is 4 μL, the catalytic current of lactate reaches the maximum. The results may be explained as follows: when there are too many enzymes on the surface of the electrode, the enzyme close to the electrode cannot contact the substrate lactate molecule. In addition, the electron transfer rate becomes worse when the active center of the enzyme is far from the electrode surface. Accordingly, the dosage of TTF was optimized (Appendix A). Therefore, the subsequent electrochemical measurements of the bioanode were carried out under the optimized conditions.

Figure 3c shows the catalytic activity of the LOx/TTF/GNRs/GCE bioanode towards its substrate lactate. The catalytic current increases with the increase in the concentration. However, when the lactate concentration is increased to 12 mM, the catalytic current is almost saturated (Figure 3c). This result demonstrates the catalytic specificity of LOx to lactate and the dependence of the lactate oxidation current on the lactate concentration. Note that the initial oxidation potential of the substrate is −0.02 V (vs. Ag/AgCl), which is lower than the redox potential of the mediator TTF (0.1 V vs. Ag/AgCl). A possible reason is that, in the absence of the substrate (lactate), the redox reaction originates from the mediator TTF within the LOx/TTF/GNRs/GCE bioanode, which requires relatively high overpotential for driving the electrochemical redox reaction of TTF. While in the presence of the lactate, LOx catalyzes the oxidation of the lactate and transfers the electrons to the TTF, leading to the accumulation of a large number of reduced TTF molecules. During positive scanning, these reduced TTF molecules with high electron energy are more easily oxidized than those in the absence of lactate. In other words, the required overpotential for the oxidation of the reduced TTF molecules is lower than that in the absence of lactate. Therefore, the onset potential could occur at a lower potential than the redox potential of the mediator TTF. Such a low initial potential for lactate oxidation reflects the effective electron-donor-acceptor within the LOx/TTF/GNRs/GCE architecture, thereby effectively promoting the electron shuttle between the active center of the enzyme and the electrode surface. In addition, chronoamperometry (CA) was performed with different concentrations of lactic acid. As shown in Appendix A, the steady-state currents increase with the increase in lactic acid and are directly proportional to the lactic acid concentration in the range of 0–20 mM. The CA currents are almost unchanged when the glucose concentration is over 19 mM, indicating the saturated catalytic capacity of the bioanode.

The long-term stability of the bioanode is important for application in EBFCs. The stability is mainly hindered by the fragile nature of enzyme molecules and the leakage of solidified chemicals (such as enzymes and mediators) [34]. Figure 3d shows the change in the anode catalytic current over 4 days. The anode catalytic current decreases significantly in the last two days. In the next few days, the catalytic current of the anode does not change significantly (Figure 3d). After 4 days, the catalytic current of the bioanode remains at ca. 50% of the initial current output. The instability of the bioanode is mainly ascribed to the following two reasons. (1) The leakage of enzymes and the mediator TTF from electrode. It can be observed in Appendix A that partial TTF leaks into the electrolyte, resulting in decreased current density. To further identify whether the enzyme and TTF are leaked into the electrolyte, the cycled electrolyte was investigated using the UV-vis diffuse absorption spectrum. As shown in Appendix A, the absorbance occurs in the 250~300 nm and 350~400 nm ranges for the cycled electrolyte solution, suggesting that both TTF and LOx are leaked into the electrolyte during the operation process. (2) The self-degradation of enzymes [35]. In the long-term operation process, the possibility of the self-degradation of enzymes will be aggravated, leading to a reduced number of active enzymes. These factors lead to the instability of the bioanode.

To verify the importance of the protective layer on the BOD/ABTS/GNRs/GCE biocathode, CV was performed on the BOD/ABTS/GNRs/GCE biocathode in the presence and absence of the protective layer. As shown in Appendix A, the redox peak current maintains 82.5% of its initial value after 200 cycles in the absence of Nafion, while the peak current of the biocathode shows negligible changes after 200 cycles in the presence of Nafion (Appendix A). The results suggest that the use of defect-enriched GNRs can significantly enhance the stability of the biocathode, which is beneficial to improving the operation stability of the biofuel cell.

Figure 3e shows the catalytic oxygen reduction reaction (ORR) of BOD/ABTS/GNRs/GCE. In the case of N_2_-saturated buffer, a pair of obvious redox peaks are detected, which is a typical redox characteristic of ABTS [28], indicating that the mediator ABTS is successfully fixed on the Appendix A. A larger ORR current starting at 0.55 V is observed in the O_2_-saturated buffer, the ORR current increases rapidly during the negative potential scan, which indicates that ABTS is an effective electron transfer mediator. The DET capability of BOD is also observed in the absence of ABTS, as shown in Appendix A. The catalytic ORR current generated by MET is significantly larger than that of DET at a potential of 0.45 V vs. Ag/AgCl. Moreover, the reduction wave observed in MET shows faster kinetics than that of DET. Therefore, the dual-immobilization of ABTS and BOD on the GNRs can result in a high ORR current and fast reaction kinetics, which makes such an immobilization strategy promising for fabricating a membranes-less biofuel cell. Appendix A shows CVs of BOD/GNRs/ABTS/GCE biocathode in 0.1 M PBS buffer (pH 7.0) with continuous N_2_-bubbling and O_2_-bubbling. The ORR current under continuous O_2_-bubbling is significantly higher than that under the quiescent condition (Figure 3e), indicating that the cathodic ORR performance is mainly limited by the slow mass transfer of O_2_.

Figure 3f shows the chronoamperometry curve of the BOD/ABTS/GNRs/GCE biocathode. The current density remains basically unchanged after 12 h of operation, indicating the formation of a stable biocathode. In addition, to verify whether ABTS is leaked from the electrode, CV was performed on the buffer after the stability measurement. As shown in Appendix A, no redox waves belonging to the mediator ABTS can be detected. On the contrary, a pair of redox peaks are detected in 0.1 M PBS (pH 7.2) containing 5 mM ABTS (Appendix A). This result shows that the immobilization of ABTS on GNRs is successful and that the biocathode has excellent durability.

### 3.3. Evaluation of the Membrane-Less Lactate/O_2_ EBFCs

On the basis of the dual-immobilization strategy, a membrane-less lactate/O_2_ EBFC was assembled using the LOx/TTF/GNRs bioanode and BOD/ABTS/GNRs/GCE biocathode (Figure 4a). The polarization and power output curves of the assembled membrane-less lactate/O_2_ EBFCs are shown in Figure 4b. The EBFC delivers an OCV of 0.62 V, with a maximum power density of 70.7 μW/cm^2^, in PBS containing 3 mM lactate. When the artificial tear is used as fuel, the EBFC exhibits an OCV of 0.58 V, with a maximum power density of 18.6 μW/cm^2^ (Figure 4c), which is better than that of the previously reported lactate/O_2_ biofuel cells using artificial tear as fuel, with the same concentration of lactate (Appendix A). The large noise in Figure 4c may be caused by the unstable mass transport of the substrates (lactate or O_2_) due to the continuous O_2_-bubbling condition. The stability of the two types of lactate/O_2_ EBFCs was evaluated by galvanostatic discharge. As shown in Figure 4d, the discharge time of the lactate/O_2_ EBFC operated in PBS and artificial tears reaches 7 h and 2.3 h at a current density of 0.02 mA/cm^2^, respectively. The discharge time of the lactate/O_2_ EBFC is relatively low when operated in an artificial tear. A possible reason is that the other substances (L-ascorbate, urea et al.) present in the artificial tear could interfere with the catalytic performance of the bioelectrodes. However, the stability is acceptable for single-use portable EBFCs because they do not require stable and long-lived power generation.

## 4. Conclusions

In summary, three-dimensional defect-enriched GNRs frameworks are rationally designed and synthesized by longitudinally unzipping the sidewalls of multi-walled CNTs, followed by hydrothermal N-doping and thermal annealing. The obtained GNRs are used as electrode materials for the dual-immobilization of enzymes and mediators for EBFC applications, showing an excellent electrochemical performance compared to that of conventional graphene and CNTs. Additionally, DFT calculations reveal the presence of a strong adsorption interaction between the defective carbon and the mediator TTF, which is responsible for the enhanced stability. The assembled membrane-less lactate/O_2_ EBFCs exhibit a high OCV of 0.62 V and a maximum power density of 70.7 μW/cm^2^ in PBS. When using artificial tear as fuel, the EBFCs can deliver a high OCV of 0.58 V and a maximum power density of 18.6 μW/cm^2^, which is also the highest performance among lactate/O_2_ EBFCs using an artificial tear as fuel (Figure 5). This work offers a design principle of electrode materials in the field of EBFCs and provides a dual-immobilization strategy for fabricating a high-performance biological electrode for bioelectronics applications, which holds a great indication in wearable devices.

## Figures and Tables

**Figure 1 nanomaterials-13-01089-f001:**
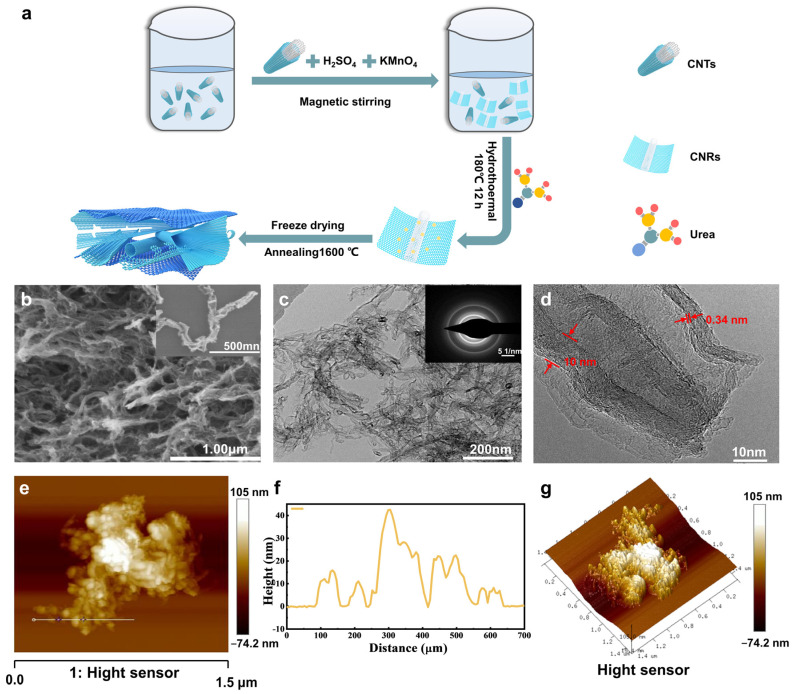
(**a**) Schematic diagram for the preparation of the GNRs composite. (**b**) SEM image of GNRs. The inset in (**b**) shows the SEM image of GNRs at a different scale. (**c**) TEM image of the GNRs composite. The inset in (**c**) shows the SAED pattern of GNRs. (**d**) HRTEM image of GNRs. (**e**) AFM image of GNRs. (**f**) Height profile of the trace line in (**e**). (**g**) 3D AFM pattern of the piece of the GNRs composite.

**Figure 2 nanomaterials-13-01089-f002:**
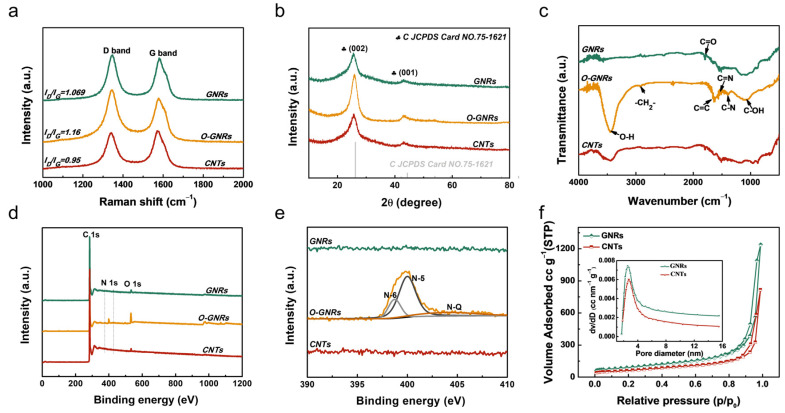
(**a**) Raman and (**b**) XRD spectra of the CNTs, O-GNRs, and GNRs. (**c**) FI-IR spectra of CNTs, O-GNRs, and GNRs. (**d**) XPS survey spectra and (**e**) N 1s spectra of the CNTs, O-GNRs, and GNRs samples. (**f**) N_2_ isotherm adsorption–desorption curves of CNTs and GNRs inset shows the corresponding BJH pore size distribution curves.

**Figure 3 nanomaterials-13-01089-f003:**
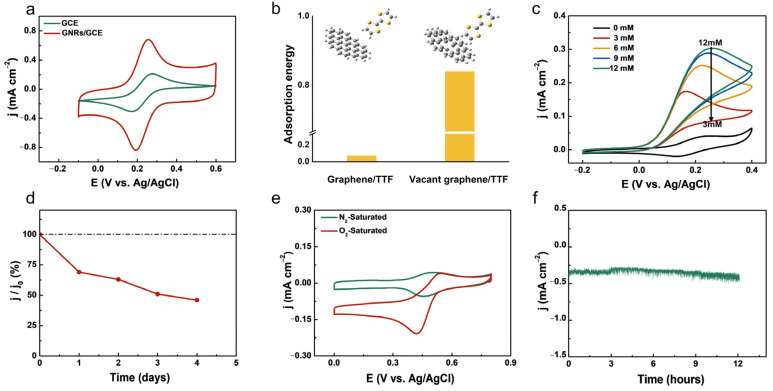
(**a**) CVs curve of GCE and GNRs/GCE in 0.1 M KCl containing 5 mM [Fe (CN)_6_]^3−/4−^. (**b**) The optimized structures of pristine graphene and vacant graphene adsorbing TTF. (**c**) CV curves of LOx/TTF/GNRs/GCE in 0.1 M PBS (pH 7.2) containing different concentration of lactate (LA). (**d**) The current density J is plotted as a function of time, and J_0_ is defined as the current density on the first day. (**e**) CVs curves of BOD/ABTS/GNRs biocathode in 0.1 M PBS solution (pH 7.2) under different atmospheres. (**f**) Chronoamperometry curve of BOD/ABTS/GNRs in O_2_-saturated 0.1 M PBS solution (pH 7.2) at a bias of 0 V vs. Ag/AgCl.

**Figure 4 nanomaterials-13-01089-f004:**
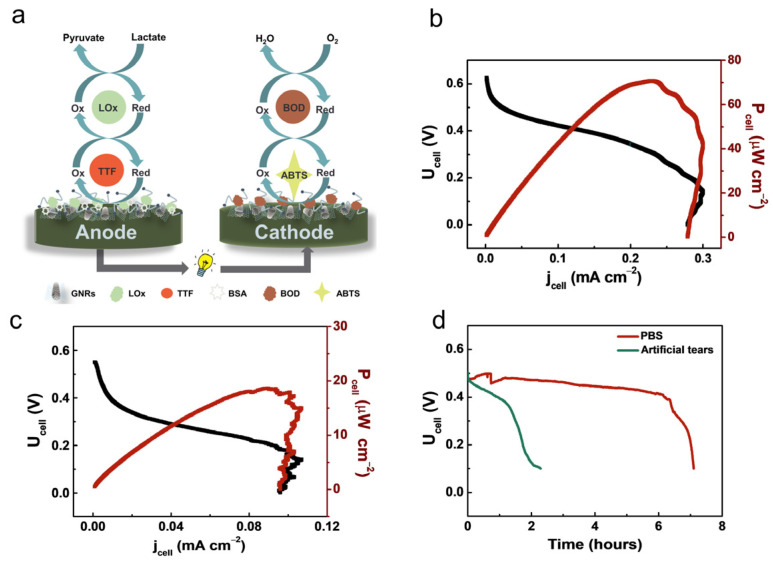
(**a**) Schematic diagram of the lactate/O_2_ EBFC. LSV and power output curves of membrane-less lactate/O_2_ EBFCs based on LOx-modified bioanode and BOD-modified biocathode in (**b**) 0.1 M PBS solution and (**c**) artificial tears (The red curve corresponds to the cell power density and the black curve corresponds to the cell voltage). (**d**) Galvanostatic discharge curves of the membrane-less lactate/O_2_ EBFCs in 0.1 M PBS and artificial tear.

**Figure 5 nanomaterials-13-01089-f005:**
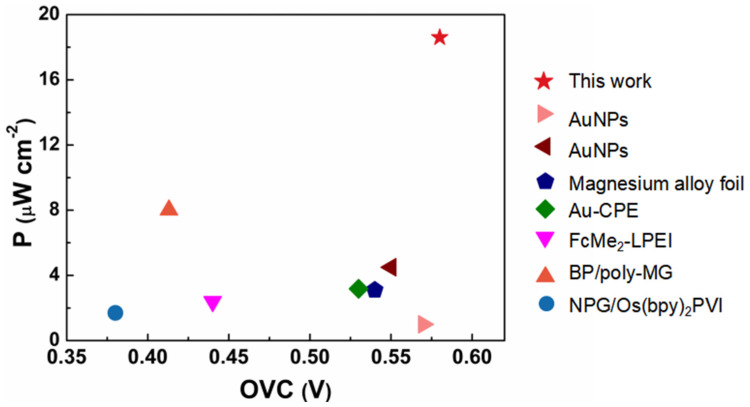
Comparison of tear-based battery performance over the years based on the information in Appendix A [36,37,38,39,40,41,42].

## Data Availability

The data presented in this study are available on request from the corresponding author.

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
