# Peer review of "Defect-Enriched Graphene Nanoribbons Tune the Adsorption Behavior of the Mediator to Boost the Lactate/Oxygen Biofuel Cell"

_nanomaterials, 2023, doi:10.3390/nano13061089_

Round 1

Reviewer 1 Report

In the presented work, an original method for the synthesis of graphene nanoribbons (GNR) with a high concentration of defects from MWCNT is proposed. It has been shown that the obtained material is capable of sufficiently firmly stabilizing adsorbed organic molecules and enzymes involved in the work of Enzymatic biofuel cells. The article quite convincingly shows that the use of GNR as a material for electrodes makes it possible to create a fairly stable and efficient biofuel cell.

The article quite fully describes the research methods used in the work and the results obtained. However, a number of presented data require clarification:

1. The scale of SEM and TEM images in Figure S1 differs by only 2 times. What is the reason for such strong differences in the image of the same material (CNT) in these images?

2. Fig.S1 and Fig.1 show SEM and TEM images of only the original sample (CNT) and the final product (GNR). O-GNR sample data not available. At the same time, data from other methods (Raman, XRD, XPS) about this sample are presented and discussed further in the text. It would make sense to include micrographs of the O-GNR sample in the work.

3. The GNR sample was heated at 1600 C. What could be the reason for the presence of hydroxyl and carboxyl groups for it (Figure S2f) after such treatment?

4. In the caption to figure S2, the samples are designated as CNT, O-GNR, GNR, and in the figures themselves the designations CNT, O-CNR, CNR are indicated. This inconsistency needs to be corrected.

5. It would make sense to present the structure of defects in GNT used in DFT calculations.

After the noted remarks are eliminated, the article can be published.

Reviewer 2 Report

Feng et al. reported the Defect-Enriched Graphene Nanoribbons tunes the adsorption behavior of mediator to boost Lactate/Oxygen Biofuel Cell. The authors have characterized the sample extensively to achieve desired properties. However, a few more clarifications are needed before the manuscript can be considered suitable for publication. A list of other comments that need to be addressed follows:

1.     An introduction is poorly written; please include the scope and limitations of the Graphene Nanoribbons, their advantages, and disadvantages. How is it better than other 2D materials?

2.     Abstract: important information missing like key results

3.     Please include the XRD plot with JCPDS card number and hkl parameters

4.     Please include the structure of the device and experimental setup.

5.     I am always skeptical when authors give linear ranges that extend several orders of magnitude. This may lead to rough estimations, not precise calculations, especially at the lower concentration levels. The simple least squares method assumes that all the signal response values have equal variances. Larger deviations at larger concentrations tend to influence the regression line more than smaller deviations associated with smaller concentrations (heteroscedasticity) leading to the inaccuracy in the lower end of the calibration range. It is better to use a lower linear range in order to avoid errors rather than give a calibration curve that spans over 3 or more orders of magnitude.

6.     The first paragraph contains trivial statements. The introduction should be reduced in length and have a focus on current analytical challenges. Essential related works can be cited.

7.     The quality of some figures is inferior and needs to be enhanced.

8.     It is better to check and correct the font size of the x and y-axis of all figures in the manuscript. It should be the same.

Reviewer 3 Report

The paper is well written and provides good results. However some improvements are necessary before publication:

- The authors need to provide some more details on the type of CNT they used

- Where were the glassy carbon electrodes purchased?

These previous details are important for the reproducibility of the results. 

- Give more details on the DFT calculations: how many atoms were used? how many defects were introduced? Of which type? At present there is very few information about the computational part

- Can the authors explain the existence of the maximum of the catalytic current for the specific amount of enzyme they found?

- The text would be more informative and sound with the addition of the comparison of the performances of the presently proposed cell with some more common cell in order to show its superiority. It is reported in the SI, but some sentences and numbers would be beneficial also in the main text

- In the Supporting information, the quality of the figures should be improved

Reviewer 4 Report

The manuscript by Li, Hu, and co-workers reports the fabrication of a glassy carbon electrode loaded with graphene, lactate oxidase, and tetrathiafulvalene for a lactate/O2 biofuel cell. It has been shown that preliminary electrochemical activation of the glassy carbon support in 0.1 M KCl, 5 mM [Fe(CN)6]3/4- aqueous solution leads to a significant increase in bioelectrocatalytic characteristics and operational stability. The topic of the study is relevant in the field and its motivation is justified. The manuscript is well structures and easy to read.

However, it has one important shortcoming. The results of the DFT calculations presented on pages 7 and 8 cannot be understood because the authors do not explain either the nature of the defects introduced into the model structure or the calculation method. What kind of defect and why was this particular type of possible types of defects chosen for analysis? Has the influence of the solvent been taken into account?

Round 2

Reviewer 3 Report

The paper has been improved. However, some answers to my questions werer properly addressed in the reply to my comments, but were not implemented in the text of the paper. I recommend to add in the paper the following information:

1) the source of the Glassy carbon electrodes (in the main text);

2) a good figure of the structure of graphene and defective graphene used for the calculations (I would place this in the supporting information; it is true there is such a graph in the present fig. 3b, but honestly it is almost impossible to understand the differences between the two structures);

3) some of the characterizations of the pristine carbon nanotubes (in the supplementing information).

Round 3

Reviewer 3 Report

THe paper was sufficiently improved and can be publish in the present form.